# Gestational Alloimune Liver Disease—Case Report

**DOI:** 10.3390/children10010066

**Published:** 2022-12-28

**Authors:** Mihaela Demetrian, Radu Botezatu, Nicolae Gică, Valentina Safta, Georgeta Grecu, Vlad Dima, Andreea Daniela Binișor, Anca Panaitescu

**Affiliations:** 1Filantropia Clinical Hospital, 011132 Bucharest, Romania; 2The Obstetrics and Gynecology Department, University of Medicine and Pharmacy Carol Davila, 050474 Bucharest, Romania; 3Emergency Clinical Pediatric Hospital “Marie S. Curie”, 077120 Bucharest, Romania

**Keywords:** alloimmune neonatal hepatitis, acute hepatic failure, neonatal hemochromatosis immunoglobulin therapy

## Abstract

We describe the case of a newborn with the antenatal onset of hepatic failure, which has been investigated for all etiologies that can cause liver damage: infectious, metabolic, genetic, and immune. The lack of a clear answer regarding the etiology and the response to immunoglobulin therapy led us to the diagnosis of gestational alloimmune liver disease. Gestational alloimunne liver disease is an uncommon and very severe cause of neonatal acute liver failure (NALF). Initially, the therapeutic approach aimed at correcting the effects produced by iron loading, respectively, iron chelators and antioxidants. Since all aspects of this case indicated characteristic features typical for GALD, therapy with intravenous immunoglobulins (IVIG) was introduced. If such therapy alters the prognosis of newborns with GALD, the etiology and pathophysiology remain uncertain. However, in cases regarding severe hepatic failure with the perinatal onset and apparently unknown etiology, immunoglobulin or exchange transfusion therapy should be taken into account even before finalizing all the etiological investigations. The prognosis is uncertain and varies between clinical resolution, chronic hepatitis/cirrhosis, and the need for a hepatic transplant, and overall survival depends on prompt therapeutic intervention.

## 1. Introduction

Gestational alloimmune liver disease (GALD) is currently considered to be the main etiological factor of liver insufficiency and neonatal hemochromatosis (NH) [1]. Affected patients mostly present a clinical picture of congenital liver cirrhosis, while some develop a hyperacute process of liver failure leading to death in utero or perinatally [2]. Most patients develop signs of liver failure immediately after birth. Laboratory findings may show hyperbilirubinemia, coagulation disorders, hepatic cytolysis, and increased serum alpha-fetoprotein and ferritin values.

Histopathological and imaging studies through salivary gland biopsy and neonatal MRI are needed to confirm extrahepatic siderosis, which is fundamental for the classic diagnosis of neonatal hemochromatosis [2]. In addition to the classical aspect, cases without iron overload have been reported and new pathological features have extended the GALD spectrum [2,3].

The pathophysiology of the disease is incomplete and unknown and a causal antigen has not been identified. The process begins early in the 12th week of pregnancy when the transplacental passage of specific immunoglobulins that activates the fetal complement system occurs and triggers the cascade of fetal liver damage [4].

The high recurrence in the following pregnancies also reflects the successive impact of maternal alloimmunization. According to some authors, C5b9 expression is more than 75% of liver parenchyma which is specific for GALD but is present in other diseases involving heavy liver inflammation and fibrosis. Liver biopsy is an important tool for diagnostics, especially in cases without hemosiderosis (NH), because is might be otherwise missed.

The treatment has evolved in the last two decades to reflect the alloimmunization perspective [4,5,6,7]. Intravenous immunoglobulin (IVIG) and exchange transfusion (ET) have replaced conventional therapy [1,2]. The exchange transfusion flushes reactive antibodies present in the bloodstream and the immunoglobulins block the complement-activating antibodies. The administration of IVIG during pregnancy (between 18 weeks of gestation and term) also potentially prevents its recurrence in the next pregnancy [2,8,9].

## 2. Case Presentation

The patient was a male term newborn (gestational age 37 weeks and 4 days), birth weight 3000 g, from a 34-year-old mother IV G, II P (the first child clinically healthy) BIII blood group Rh-positive. Previously, the mother experienced two early miscarriages. The evolution of the currently investigated pregnancy was complicated by gestational hypertension (onset at 35 weeks) without other pathological elements: negative screening for chromosomal abnormalities, normal second-trimester scan, negative TORCH serologies, and absent group B streptococcus. She had no history of drug or medicine abuse. The mother was admitted to advanced labor. At admission, the cardiotocographic aspect (CTG) showed the lack of variability of the fetal heart rate and of the sleep-wake cycle (Figure 1). The birth was spontaneous in cranial presentation, there was green amniotic fluid at the rupture of the membranes, a thin umbilical cord, an Apgar score of 2/4/6 and required resuscitation and intubation maneuvers immediately after birth.

The newborn was admitted to the NICU with an extremely severe condition, pale skin, generalized edema with the appearance of hydrops, generalized purpura, ecchymosis of the cephalic extremity, parietooccipital cephalohematoma, abdomen with important wall edema and ascites fluid. Thoracoabdominal radiography and abdominal ultrasound showed the presence of ascites fluid and poor aeration of the colic frame (Figure 2).

The newborn exhibited hepatomegaly with the right lobe extending caudally. Enlarged hepatic vein and associated splenomegaly were observed on abdominal ultrasound. A cardiac ultrasound revealed mild pulmonary hypertension, mild sized patent ductus arteriosus with left to right shunt, patent foramen ovale with left to right shunt, representing mild cardiac involvement which responded accordingly to the inotrope and vasopressors agents.

A paracentesis was performed and approximately 150 mL of serous citrine ascites fluid was extracted.

Biological samples revealed: hyperleukocytosis (44,500/mmc) with neutrophils (35%) and lymphocytosis (57%), thrombocytopenia (88,000/mmc), mild anemia (Hb = 10.9 mg/dL), mixed acidosis (PH = 7.13, pCO2-53, PO2-71.9, BE = −11.4) Na 132.7 mmol/L, BIII blood group, Rh positive, negative Coombs test (excluding Rh alloimmunization/maternal-fetal group), altered coagulation: INR increased, PTT increased, AP below 60%, low fibrinogen and increased D-Dimers.

To exclude the infectious etiology, we collected blood culture, CSF culture, immunology profile TORCH (MINI VIDAS): IgG and IgM and DNA for Cytomegalovirus, Rubella, Toxoplasma, RBW-TPHA, Herpes antibodies, serology for Listeria, Parvovirus and hepatitis C and B. All results were negative [10]. Metabolic (carbohydrate, amino acid and lipids) and endocrine abnormalities were excluded by extensive metabolic testing (47 abnormalities including phenylketonuria and hypothyroidism), without detecting any abnormalities [11]. 

Causes of autoimmune hepatitis transmitted via maternal diseases (smooth muscle antibodies (ASMA), anti-LKM1 antibodies and antinuclear antibodies) were also verified and no autoimmune immunological conflict was identified. Dynamic paraclinical investigations revealed the appearance of cholestatic neonatal hepatitis: increased liver enzymes (ALT, AST), hyperbilirubinemia with the important direct component and there were increased triglycerides and cholesterol. (Table 1) The undulating but still limited nature of hepatic cytolysis and cholestasis was noted, as also the very high value of serum ferritin in the seven days of life.

The treatment consisted of the correction of coagulation disorders with fresh-frozen plasma (FFP) isogroup, isoRh, correction of anemia with packed red blood cells, correction of metabolic disorders and inotropic and vasoactive support. We focused more on the most common causes of neonatal liver failure, especially the infectious ones. Ampicillin and Gentamicin were administered and hemodynamic support and correction of hematological disorders were provided. At that time, with an uncertain etiology and pathophysiology, intravenous immunoglobulin seemed to be the most appropriate choice.Two doses of intravenous immunoglobulins (2.5 g) were also administered at 7 and 14 days of life, respectively. IVIG was administered when we had a strong suspicion of alloimmune disease, but it was to late to exchange transfusion.

On the third day of life, the patient presented clinically and electrically generalized tonic-clonic seizures (aEEG monitoring)—requiring administration of Phenytoin (i.v) and Midazolam until the 8th day, then continued with Phenobarbital per os. A cranial ultrasound performed on the second day of life showed a round-oval hyperechoic image of 0.6/0.4 cm in the relatively superficial left frontal-parietal cerebral parenchyma which evolved to a left periventricular leukomalacia and left porencephalic cyst at one month of age (Figure 3 and Figure 4). The edema gradually reduced and the ascites disappeared; the intestinal transit became functional after 3–4 days and the enteral feeding was initiated. Parenteral nutrition was discontinued after 14 days. Neurological examination at one month of age accompanied by an EEG recording did not show any significant abnormalities. The patient was discharged after 40 days of hospitalization. At 2 months of evaluation, the patient showed an ascending weight curve and significant improvement in cytolysis and hepatic cholestasis.

## 3. Discussion

Neonatal liver failure represents a major emergency due to the risk of death and late complications such as liver cirrhosis. This type of pathology with a very severe clinical picture poses great difficulty for the clinician because prompt diagnosis is the key to a higher survival rate.

The diagnosis of congenital alloimmune hepatic impairment in our case was one of exclusion. Antenatal damage has been evident since the mother was admitted when cardiotocographic monitoring showed signs of fetal distress. The CTG monitoring showed a lack of variability in fetal heart rate (an element that may be the result of hypoxic damage to the fetal central nervous system by influencing the interaction between the sympathetic and parasympathetic nervous systems) [12]. Another element observed was the lack of a sleep cycle (“cycling”) that can hide a deficiency of the central nervous system regarding the regulation of heart rate [13].

Given that the mother was hospitalized shortly before birth, the appearance of CTG can also be considered as part of gradual hypoxia in the decompensated phase, when the variability disappears and the baseline decreases [14].

After analyzing all the causes of cholestatic hepatitis, when the patient was very severely affected and required urgent measures, we quickly eliminated the infectious causes in an early stage (early neonatal sepsis and congenital infections). HSV infection was excluded from the diagnosis by performing the TORCH panel immediately after birth. The newborn was extracted by cesarean section with no skin lesions suggestive for HSV infection. Therefore, there was no specific symptomatology for antenatal herpes, and the first trimester serologies were negative. In our hospital, we have had no confirmed cases of neonatal herpes virus infection in the last 10 years. The clinical picture was dominated by coagulation disorders, as evidenced by the appearance of hemorrhage in the cerebral parenchyma in the first 24 h of life which clinically manifested with convulsions. Therefore, the correction of coagulation disorders by FFP transfusions (a total of 5 FFP transfusions in the first 3 days) was primordial in the therapeutic approach. Immunoglobulin therapy was applied after the first 7 days of life and its effect was beneficial, noted by the normalization of coagulation samples, increased platelet count and self-limitation of the process of hepatic cytolysis. Although we did not perform a salivary gland biopsy and did not prove extrahepatic siderosis by MRI, the clinical picture of hepatic failure and high level of ferritin made the diagnosis highly suspicious of neonatal hemochromatosis following alloimmune liver disease.

Although the liver damage was antenatal, we were afraid to perform liver biopsy and we missed the histopathological evidence of cirrhosis. At first, the clinical condition and coagulation disorders did not allow for liver biopsy, and the subsequent evolution did not require an invasive intervention for diagnostic purposes only.

The etiology and pathophysiology of neonatal hemochromatosis(NH) still remain unknown. Iron loading in various extrahepatic tissues in neonates with liver disease was thought to be caused by genetic errors in iron metabolism. However, it has become clear that iron overload was not the primary mechanism of the disease, but appears to be the phenotypic result of fetal liver injury [15].

The congenital alloimmun disease involves an interaction between the immune system of the mother and the fetus that is a maternal allograft [16]. The fetus produces an alloantigen that sensitizes the mother’s immune system and produces specific reactive antibodies. Since only immunoglobulin G (IgG) can be transported across the placenta, fetal damage due to maternal alloimmunization is mediated by IgG [17,18]. The major pathology of NH is limited to the liver and most cases show extensive parenchymal lesions. It is assumed that some components of the fetal immune system are also involved because the mere binding of IgG to a cell should not cause such damage [19].

Evidence suggests that most cases of neonatal hemochromatosis are the phenotypic expression of fetal liver damage due to alloimmunization. Alloimmune diseases involve the transplacental passage of specific G immunoglobulins that activate the classical fetal complement pathway. The activation of the fetal complement leads to the formation of the membrane attack complex (MAC) responsible for affecting the liver cells [20,21]. Histopathological studies on fragments of affected liver tissue demonstrated the existence of membrane complexes [21,22].

The high recurrence in the following pregnancies also reflects the successive impact of maternal alloimmunization. The clinical appearance of all forms of congenital alloimmune liver disease has a wide range of severity, which is why survival rates range from 20% to 80%. However, the risk of death is associated with delayed diagnosis [8]. In our case, earlier miscarriages were an anamnestic hint leading to the diagnosis of GALD.

Although there has been an increase in understanding of this disease, the discovery of the specific alloantigen is still needed. In the last two decades, the evolution of treatment has changed to reflect the perspective of alloimmunization; therefore, intravenous immunoglobulin and exchange transfusion have replaced conventional therapy and markedly improved the prognosis.

Similar to other known alloimmune disorders such as alloimmune thrombocytopenia, a woman may have several normal pregnancies, but once pregnancy was affected there is a high risk, as high as 90%, of recurrence in the following pregnancies. The weekly administration of IV IG starting in the 18th week of pregnancy until the end of gestation potentially prevents its recurrence in the next pregnancy and showed excellent outcomes [23].

Prenatal diagnosis of GALD with subsequent NH might be suspected based on fetal intrauterine growth restriction, oligohydramnios, ascites, placental hydrops or splenomegaly with negative workup for infectious causes. Fetal MRI might be helpful in detecting liver atrophy or iron overload in hepatic or extrahepatic tissue, although it is not a constant finding [24,25].

## 4. Conclusions

All the aspects of this case indicate typical and possible aspects or features of GALD although we did not use classic diagnostic tools (MRI or mucosal biopsy or liver biopsy). Congenital alloimmune hepatitis should be considered in the presence of a clinical picture of acute hepatitis with cholestasis, jaundice and coagulopathy after the rapid elimination of infectious and metabolic causes. Administration of IVIG and exchange transfusion should not be delayed. It may have been safer and faster to exclude HSV infections via PCR and immediately start empiric Acyclovir therapy pending results, because herpes virus infection is listed as the second cause of NALF. Unfortunately, the indications for empiric acyclovir are not yet standardized.

The pathology presented was new to us and represented a great challenge. Going through the diagnostic steps was chronophagous and waiting for investigation results delayed IVIG therapy. Performing an exchange transfusion in the first days of life would certainly have had a benefit and would have shortened the length of hospital stay.

## Figures and Tables

**Figure 1 children-10-00066-f001:**
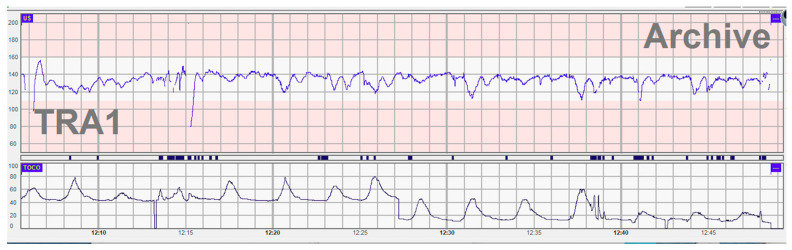
The cardiotocographic aspect during labor.

**Figure 2 children-10-00066-f002:**
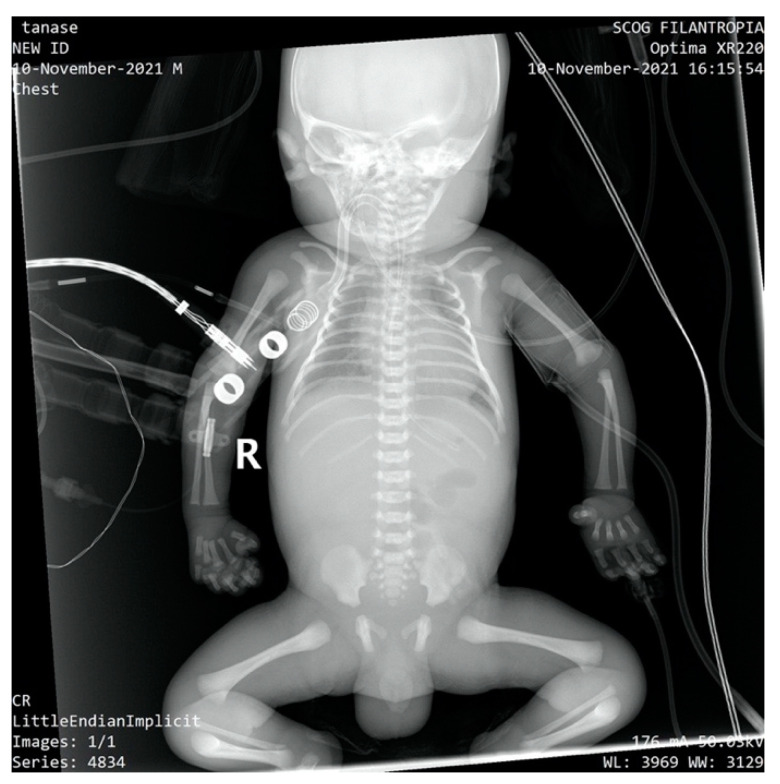
Thoraco-abdominal X-ray-poor aeration of the colic frame, ascites fluid.

**Figure 3 children-10-00066-f003:**
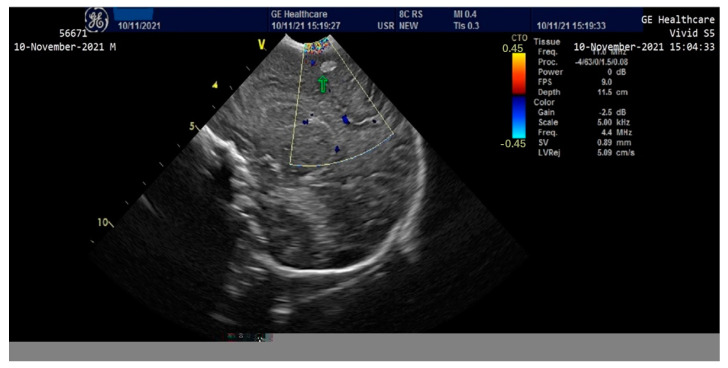
Cranial ultrasound in the first 24 h of life with round-oval hyperechoic image 0.6/0.4 cm in the left frontoparietal cerebral parenchyma.

**Figure 4 children-10-00066-f004:**
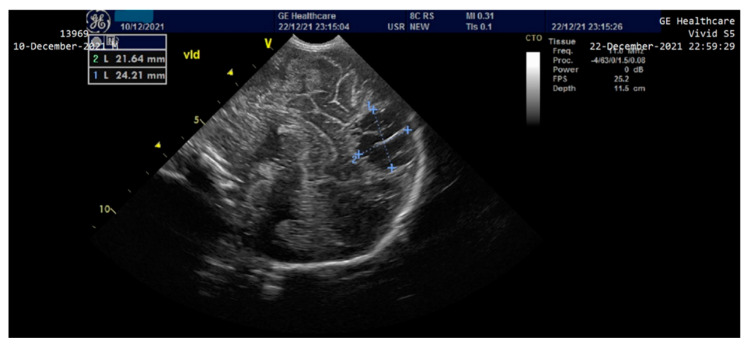
Cranial ultrasound at one month of age—Appearance of left periventricular leukomalacia, left porencephalic cyst.

**Table 1 children-10-00066-t001:** The main laboratory investigations.

Laboratory Investigation	Day 1	Day 2	Day 3	Day 7	Day 14	1 Month	2 Months
Hemoglobin (g/dL)	10.9/35	11.4/34.5	13/40	13.4/42	12/36	12/36	10.5/29
Leukocytes/mm^3^	44.500	18.400	32.000	22.700	6200	12.300	10.000
Platelets/mm^3^	88.000	28.000	86.000	66.000	235.000	635.000	520.000
PTT (s)	60	54	38	28	26	27	-
AP (%)	49	65	72	80	80	80	-
INR	2.65	1.2	1.2	1.1	1.2	1.2	-
D-Dimers	18.2	19.3	12	3.5	1.7	0.3	-
Fibrinogen mg/dL	34	45	86	136	145	381	-
AST (UI)	702	462	150	696	367	256	106
ALT (UI)	292	240	66	432	223	186	117
GammaGT (U/L)				535			120
Serum ferritin (ng/mL)				1556			275
Bilirubinemia (T/D) mg/dL			12/3	18/5.8	12/3.5	12/3.7	2.74/1.4
Triglycerides/Chol mg/dL		288/255	305/208	245/134	165/122	280/170	108/138
ALP (U/L)				122	420	432	483
Interleukin 6 (pg/mL)		161					
CRP/Procalcitonin mg/L/ng/mL	70/2.93	120/18	30	8	<4	<4	0.33
Alpha 1 AT (g/L)		1.46					
Alpha FP (ng/mL)				>60.500			

PTT—Partial Thromboplastin Time, INR—international normalized ratio, AST—aspartate aminotransferase, ALT—alanine aminotransferase, ALP—alkaline phosphatase, CRP—C reactive protein, AT—antitrypsin, FP—fetoprotein.

## Data Availability

No new data were created or analyzed in this study. Data sharing is not applicable to this article.

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
