# Peer review of "Gestational Alloimune Liver Disease—Case Report"

_children, 2022, doi:10.3390/children10010066_

Round 1
Reviewer 1 Report
This is an extremely rare case report about alloimmune neonatal hepatic failure. This case review deserves to be considered a useful reference for institutions who do not familiar with this rare disease.
Author Response
Thank you for reviewing our article!
We are glad that this paper meets your expectations and we hope we will be able to provide satisfying answers to all our reviewers.
Reviewer 2 Report
The authors present a case of neonatal liver failure, probably caused by alloimmune liver disease. The case does not seem typical for gestational alloimmune liver disease (GALD), as for example the aminotransferases were higher as to expect in typical presentation and no verification of iron deposition or liver fibrosis/cirrhosis is available (see comments below).
I have a few questions/comments:
1. Was hyperammonemia present?
2. Was transferrin/transferrin saturation measured?
3. The authors discuss that they did not perform any studies to examine (excess) iron deposition. The presentation seems not typical for GALD-NH. The increased ferritin may perhaps be more contributed to the hepatocellular injury than to real iron accumulation (as indicated by the normalised ferritin concentration at 2 months without any iron chelating therapy). Do the authors have any iron quantification, for example by MRI during the further course?
4. The more common form of GALD presents with neonatal haemochromatosis and liver cirrhosis. Do the authors have any information on fibrosis/cirrhosis of the patients liver? As the patients seems to recover during the further course, one may speculate that he may not have (advanced) cirrhosis?
Taken together comments 3 and 4 the actual case may be a more rare presentation of GALD as acute liver failure without neonatal hemochromatosis and cirrhosis? Perhaps the authors could comment/discuss this.
5. IVIG was first given at day 7. I assume it was not given earlier as the diagnosis was unclear before?
Author Response
Hi!
Here you can find attached our answers.
Thank you for reviewing our article!

Reviewer 3 Report
Mihaela et al. reported a case that due to the clinical picture and due to the response followed by performing an empiric therapy suffered from GALD.
While the discussion of the disease and of the clinical disease cause is scientifically sound there are some issues to point out:
1) Titel: “alloimmune neonatale hepatic failure” is not the common name of the disorder, Gestational alloimune liver disease would fit better
2) abstract:
- Line 12: Where toxic causes of NALF excluded?
3) Introduction: A diagnostic criterion for GALD is the identification of C5b9 complement deposits in a heavily inflamed and fibrotic liver. Therefore, please mention liver biopsy as an important diagnostic tool. Especially because cases without NH exist and might be missed if liver biopsy is not conducted.
4)
Case Presentation
- Line 54: Highlight earlier miscarriages as an anamnestic hint leading to the diagnosis of GALD
- Line 61/62: What could be the explanation for the thin umbilical cord?
- Line 67/68: Did the ultrasound show a patent ductus venosus? Did you see a liver fibrosis? What degree of liver fibrosis? Liver morphology should be described!
- Was an echocardiography conducted?
- Was aciclovir given until herpes simplex virus infection was excluded? Why was it not excluded using PCR? This is recommended as has higher sensitivity, results are retreived much earlier.
- Line 86/87:„Dynamic paraclinical investigations“ - What does it stand for ? What is ment by this expression?
- Line 100: Mention does of IVIG/kg, why did you not perform exchange transfusion?
- The structure of the part “case presentation” is a little bit confusing. It Should for example be structured following this order: Anamnesis and clinical findings, then diagnosis, therapy, and course. (Diagnosis and what made you start therapy with IVIG is not mentioned in this part of the manuscript, which makes it hard to understand/follow your thoughts. Laboratory findings of liver failure are displayed in two parts of this subsection, would be better for the understanding, if mentioned together)
- -IL6 was elevated did you only measure it once? Was blood cultured? Was an antibiotic given? An infection like sepsis may have also caused this critical illness.
5) Conclusion
- Line 184: Liver cirrhosis doesn`t have to be a late complication, it can be the primary finding due to the early antenatal development of the disease (as also indicated in this case describing fetal ascites)
- Mention application of aciclovir (HSV-Infection 2nd common cause of NALF!)
- - What is new in your case? What went well, what could have been better? This is an improtant isssue, as important aspects of the diagnostic work up has not been performed. Please emphasize a bit more the important “take home message”?
-
Author Response
Thank you for reviewing our article!
You can find attached the answers.

Round 2
Reviewer 3 Report
thank you for submitting a revised version:
I have still some comments:
1) Please modify - - Line 49: not “is” but “it”
2) line 75: "Hepatomegaly with right lobe extending caudally. Enlarged hepatic vein, associated splenomegaly. There was no evidence of hepatic fibrosis on the ultrasound." How do you explain the enlargement of the hepatic veins? Is it congestion ? You may add the cardioechography to better explain of exclude causes for this ultrasonogaphic finding. It does not necessarily go along with a splenomegaly...
3) You need to shorten the conlusion and add parts of it to the discussion. I have a problem with your HSV treatment standard. It is not sufficient to wait in case of a connatal HSV infections for skin rash etc. In about one third of these cases there are no obvious clinical signs of an HSV infection. This is a highly letal disease and infectious complication, please review:
https://www.sciencedirect.com/science/article/pii/S0022347615002516
Therefore, this must be critically discussed! Use PCR in future to exclude connatal HSV infections and immediatly start Acyclovir treatment until HSV is excluded via PCR. This is a weakness of your procedure and the manuscript and you need to discuss this accordingly.
Also in lines 212/213 you write: "we had no histopathological evidence". This sentence is misleading. Please state that you refrained from performing a liver biopsy because of ...
In your final evaluation of this case you then need to summarize that all the aspects of this case are indicating typical and possible aspects or features of GALD but that you did not use proper diagnostic tools (MRI or mucosal biopsy or liver biopsy).
Kind regards
Author Response
Thank you again for the new observations made after the second revision.
We are trying to compile with all the reviewer's demands.
We hope that the latest version of the manuscript that we are loading on the MDPI platform meets all the criteria you are suggesting to us.
Round 3
Reviewer 3 Report
Thank you for the revised version, there is one aspect to modify:
Abstract, line 17: "Following the discovery of the immune mechanism, (mater- 17 nal-fetal alloimmunization)..."
It still sounds that you have made the clinical diagnosis by performing MRI, lip or liver biopsy. As there is no specific lab chemistry marker you can only suggest the diagnosis GALD. Therefore, you should modify this sentence e.g. "Since all aspects of this case were indicating characteristical features typical for GALD the therapy..."
Author Response
Merry Christmas!
Thank you for your patience in reviewing our paper again and again. We are doing our best to modify our paper according to your suggestions.
Now we made the changes you asked for in your last report.
We hope that this time our manuscript will be suitable for you to accept in its current form.
Thank you again!